



# Uncertainties in carbon emissions from land use and land cover change in Indonesia

Ida Bagus Mandhara Brasika[1,2], Pierre Friedlingstein[1,3], Stephen Sitch[4], Michael O'Sullivan[1], Maria Carolina Duran Rojas[1], Thais Michele Rosan[4], Kees Klein Goldewijk[5], Julia Pongratz[6,7], Clemens Schwingshackl[6], Louise P. Chini[8], George C. Hurtt[8]

[1] Department of Mathematics & Statistics, Faculty of Environment, Science & Economics, University of Exeter, Exeter, United Kingdom One, Institution One, City One, Country One

[2] Department of Marine Science. Faculty of Marine Science & Fisheries, University of Udayana, Bali, Indonesia

[3] Laboratoire de Météorologie Dynamique, Institut Pierre-Simon Laplace, CNRS, École Normale Supérieure, Université PSL, Sorbonne Université, École Polytechnique, Paris, France

[4] Department of Geography, Faculty of Environment, Science & Economics, University of Exeter, Exeter, United Kingdom

[5] Department IMEW. Faculty of Geosciences, Copernicus Institute of Sustainable Development, Utrecht University, Utrecht, the Netherlands

[6] Department of Geography, Ludwig-Maximilians-Universitat Munchen, Munich, Germany

[7] Max Planck Institute for Meteorology, Hamburg, Germany

[8] Department of Geographical Sciences, University of Maryland, College Park, USA

*Correspondence to*: Ida Bagus Mandhara Brasika (i.brasika@exeter.ac.uk)

**Abstract.** Indonesia is currently one of the three largest contributors of carbon emissions from land use and land cover change (LULCC) globally, together with Brazil and the Democratic Republic of the Congo. However, until recently, there was only limited reliable data available on LULCC across Indonesia, leading to a lack of agreement on drivers, magnitude, and trends in carbon emissions between different estimates. Accurate LULCC should improve robustness and reduce the uncertainties of carbon dioxide ($CO_2$) emissions from Land Use Change (ELUC) estimation. Here, we assess several cropland datasets that are used to estimate ELUC in Dynamic Global Vegetation Models (DGVMs) and Bookkeeping models (BKMs). Available cropland datasets are generally categorized as either census-based such as the Food and Agricultural Organization (FAO) annual statistical dataset, or satellite-based such as the Mapbiomas dataset, which is derived from Landsat Satellite images. Our results show that census-based and satellite-based estimates have little agreement on temporal variability and cropland area changes. In some islands, they show spatial similarity, but differences appear in the main islands such as Kalimantan, Sumatra and Java. These differences lead to spatio-temporal uncertainty in carbon emissions. The different land cover forcings (census-based vs satellite-based) in a single model (JULES-ES) result in ELUC uncertainties of about 0.08 [0.06 to 0.11] PgC/yr. Furthermore, we found that uncertainties in ELUC estimates are also due to differences in the carbon cycle models in DGVMs, as DGVMs driven by the same land cover dataset show differences in ELUC estimates of 0.12 ± 0.02 PgC/yr with 95% confidence level and range [-0.04 to 0.35] PgC/yr. This is consistent with other product such as BKMs that estimates 0.14 [0.12 to 0.15] PgC/yr with both steady trend. We also compare emissions with those from the National Greenhouse Gas



Inventory (NGHGI) product. The NGHGI estimates (based on BUR3; periodic official government report on Greenhouses Gas to UNFCCC) have much lower carbon emissions (0.06 ± 0.06 PgC/yr), though with an increasing trend. These numbers double when we include emissions from peat fire and peat drainage: the DGVM ensemble indicates emissions of 0.23 ± 0.05 PgC/yr and BKMs indicate emissions of 0.24 [0.22-0.25] PgC/yr. In contrast, emissions based on the Indonesian NGHGI remain much lower (BUR2: 0.18±0.07 PgC/yr BUR3: 0.13 ± 0.10 PgC/yr). Furthermore, emission peaks occur in year of moderate-to-strong El Nino events. Several improvements might reduce uncertainties in carbon emissions from LULCC in Indonesia, such as: combination of satellite-based dataset with census-based dataset, inclusion of peat-related emissions in DGVMs and potentially explicit inclusion of palm oil in the models as this is a major crop in Indonesia. Overall, the analysis shows that carbon emissions have no decreasing trend in Indonesia, Therefore, deforestation and forest fire prevention remain vital for Indonesia.

## 1 Introduction

Indonesia has a significant role in the global carbon cycle through carbon emissions from land use and land cover change (LULCC). Together with Brazil and the Democratic Republic of Congo, these three countries are the highest carbon emitters in the world from land use change. Their emissions combined are more than half of the global total land-use emissions (0.75 [0.59-0.87] PgC/yr or 56 [44-68]%)(Friedlingstein et al., 2023).

Indonesian forests are spread across several islands. The major islands are Sumatra, Kalimantan, Sulawesi and Papua(Gaveau et al., 2021). In recent decades, two islands, Sumatra and Kalimantan, have experienced extensive land use changes(Gaveau et al., 2014, 2016; Margono et al., 2012). There were many forested areas that have now been converted into agriculture lands, predominantly palm oil plantations. Forest area declined by 9.79 MHa or 11% between 2001 and 2019, with near equal expansion of oil palm plantation of 8.48 MHa(Gaveau et al., 2022).

These high rates of land conversion in regions of carbon-dense and often pristine natural forests result in high carbon emissions(Hong et al., 2021). Although the Indonesian government has applied several policies promoting forest conservation, e.g. granting "community forest" over 12.7 MHa to be conserved by local communities, in practice deforestation increased, opposite with the intended goal (Kraus et al., 2021). Furthermore, there are large peat and forest fires during extreme dry seasons in Indonesia(Fernandes et al., 2017; Nechita-Banda et al., 2018). This has added to the carbon released to the atmosphere from Indonesia (Brasika, 2023; Van Der Werf et al., 2017). Although there is a declining trend in reported forest fire between 2003-2018 (van Wees et al., 2021), fires continue to contribute significant emissions. For example, in 2019-2020, deforestation fires in Indonesia continued contributing to high greenhouse gases (GHG) emissions, about 3.7 ±0.4 Gt CO2eq and half of this can be attributed to emissions from peatland fires (Datta and Krishnamoorti, 2022).

The emission of carbon from LULCC caused by deliberate human activities is represented as ELUC (Emissions of Land Used Change) (PgC/yr) in the Global Carbon Budget (GCB) calculated with a set of bookkeeping models (Friedlingstein et al., 2023). This includes CO2 fluxes from deforestation, afforestation, logging, forest degradation, shifting cultivation and





regrowth of forests. ELUC can also be calculated by Dynamic Global Vegetation Models (DGVMs) which is part of Trends in Net Carbon Exchange Project (TRENDY) (Friedlingstein et al., 2019; Sitch et al., 2015), which supplies ensemble DGVM
results each year to GCB.

In GCB, DGVMs use LULCC forcing from Land Use Harmonization (LUH2) Dataset (Chini et al., 2021; Goldewijk et al., 2017; Hurtt et al., 2020). LUH2 is designed for global-scale analysis in climate models over multi-century time periods and uses the HYDE historical cropland, grazing land, and urban land areas which are largely informed over recent decades by national statistical census-based data of FAO. As a result, LUH2 can have some temporal and spatial challenges (Rosan et al.,
2021) when being employed for national-scale analysis over contemporary time-periods.

With the development of multi-year land cover datasets from Earth Observation, there is potential to reduce the uncertainty in carbon emissions from LULCC. This has previously been applied in the GCB 2022 (Friedlingstein et al., 2022) where the authors replace the FAO land cover data with satellite based Mapbiomas data for Brazil (Rosan et al., 2021). This forcing data, utilised country total cropland and grazing lands from Mapbiomas, but still used the standard spatial allocation framework
(HYDE;(Goldewijk et al., 2017)). This was to ensure backward compatibility to data prior to the satellite period. The Mapbiomas dataset is derived from Landsat Satellite Observation which has been validated through ground data sampling (Mapbiomas Indonesia - Collection).

In this research, we apply a similar approach to improve our understanding about the uncertainties of carbon emissions from LULCC in Indonesia over the last two decades (2000-2022). We used LUH2-GCB2022, which is the harmonization of several
datasets include satellite data but the cropland and grazing land from census-based dataset FAO, and satellite-based cropland (Mapbiomas Indonesia 1.0 (MB1) and Mapbiomas Indonesia 2.0 (MB2)) datasets as input to JULES-ES. First, we compare the temporal and spatial variability of cropland from these datasets. Second, we simulate ELUC with two approaches: various LULCC driver datasets (LUH2-GCB2022, MB1 and MB2) in one model JULES-ES and compare against results using one LULCC driver (MB1) in various models in TRENDYv12. Last, we compare our estimates with other models/products which
are Bookkeeping Models (BKM) and National Greenhouse Gases Inventory (NGHGI) for Indonesia.

## 2 Methods

### 2.1 Land Use Dataset

#### 2.1.1 Mapbiomas Indonesia Dataset

Mapbiomas is an initiative aimed at mapping LULCC using a fast, reliable, and cost-effective methodology. Initially launched
in Brazil in 2015, it has since been applied to other countries. Indonesia became the first non-Latin American country to adopt the method, receiving training from experts in Brazil(Experts from the Asian country were trained by the Brazilian MapBiomas team to launch the land use mapping platform; Silva et al., 2022). Mapbiomas utilizes Landsat images and employs a pixel-



based approach. Machine Learning algorithms have been developed to analyse these images, leveraging the powerful cloud processing capabilities provided by the Google Earth Engine (GEE).

The resulting dataset, Mapbiomas Indonesia 1.0 (MB1), classify the LULCC from 2000 to 2019 (Mapbiomas Indonesia - Collection). While the classification scheme is adapted from Mapbiomas Brazil, it also includes specific characteristics relevant to Indonesia. For instance, agriculture in Mapbiomas Indonesia includes a palm oil category, reflecting the significant impact of palm oil production on LULCC in Indonesia (Mapbiomas Indonesia - Collection). However, there is no specific grazing land category in Mapbiomas Indonesia, as grazing land is considerably small and steady in Indonesia. Nevertheless,

we still use the grazing land from Land Use Harmonization (LUH2- GCB2022) for completeness (Fig. 1). In case total combination of Mapbiomas agriculture fraction and LUH2- GCB2022 grazing land fraction is higher than 1 in a given grid-cell, we reduce the grazing land fraction accordingly.

Mapbiomas has high spatial resolution of 30 meters, which is significantly higher than other products such as ESA CCI LC at 300 meters, the Mapbiomas dataset has the potential to detect even small changes in LULCC. This dataset finds various

applications, including the calculation of $CO_2$ emissions resulting from land use change(Garofalo et al., 2022). Currently, Mapbiomas Indonesia released its Collection 2.0 version (MB2). This recent dataset has completed an accuracy assessment and validation process (use ~12,967 samples) with result overall accuracy 77.2%. This MB2 covers a longer period 2000-2022, with some improvement in land use classification. The agriculture is divided into more categories: Palm Oil, Rice Paddy, Pulpwood Plantation and Mosaic Agriculture.



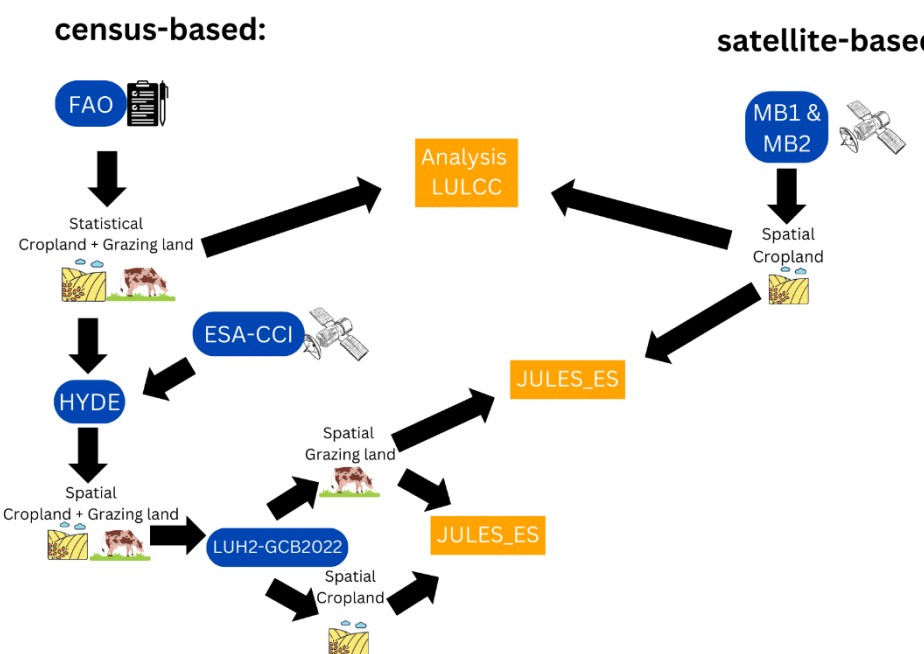

**Figure 1: The diagram of different LULCC driver dataset that is used as input for JULES-ES simulation.**


### 2.1.2 Integration Multiple Datasets (FAO, HYDE, LUH2 and Mapbiomas)

The LUH2 Dataset employs an accounting-based method to provide global annual gridded land-use states and transitions for the historical period from 850 to 2023 (Hurtt et al., 2020). LUH2 considers each grid-cell as naturally forested or non-forested, this potential vegetation area is adjusted based on information about land-use activities such as agricultural expansion and

abandonment, wood harvesting, urban development, and shifting cultivation.

LUH2 received input of the land fractions in agriculture from The History Database of the Global Environment (HYDE) [17]. This is a land use dataset that combines historical population estimates and allocation algorithms with time-dependent weighting maps to ensure internal consistency(Goldewijk et al., 2017). The cropland and grazing data in HYDE are obtained from the country-level statistical data provided by the Food and Agriculture Organization (FAO) as shown in Fig. 1.

The FAO utilised census-based approach provided by each country (Ricciardi et al., 2018). The national data are provided to FAO annually via questionaries filled by each country; however they may be based on less frequent census periods, which can vary among the countries. In Indonesia, FAO started using questionnaires to collect the data on 2016 (FAO, 2022), the data before that year is from the Indonesia Statistic Agency (BPS) who conducted agriculture surveys every 10 years (Badan Pusat



Statistik, 2023), the years in between are interpolated. To spatially allocate the FAO data, HYDE uses maps from the ESA
Land Cover Consortium (Fig. 1).

Table 1 The LULCC dataset

| | Dataset | In GCB for Indonesia | Temporal | | Spatial | DGVMs on this research |
|---|---|---|---|---|---|---|
| 1 | LUH2-GCB2022 | GCB 2022 | 2000-2022 | Extrapolated 2020 onward | Distributed by HYDE | JULES-ES |
| 2 | MB1 | GCB 2023 | 2000-2022 | Extrapolated 2019 onward | Available | Directly use in JULES-ES ; Extrapolated with HYDE and adjusted with LUH2 in TRENDY |
| 3 | MB2 | | 2000-2022 | | Available | JULES-ES |

This version of LUH2 with cropland areas based on HYDE/FAO (except Brazil that started use Mapbiomas in 2022), is utilised
by Global Carbon Budget until the version 2022 (GCB22). As the FAO dataset is only available until 2020, the years 2021
and 2022 are extrapolated from HYDE, using the trend over the previous 5 years (2016-2020). Then specifically for Indonesia,
in the Global Carbon Budget 2023 (GCB23), FAO data for Indonesia were replaced (while other countries except Brazil
continued to be based on HYDE/FAO) with the satellite-based dataset Mapbiomas Indonesia 1.0 (MB1) as input for HYDE
then used by LUH2 for the period 2000-2019, with extrapolation from HYDE until 2022. In recent year, Mapbiomas launched
the new version of Mapbiomas Indonesia 2.0 (MB2) which is claimed has higher accuracy. This dataset is currently proposed
for future GCB. Thus, in this research we differentiate the LUH2- GCB2022, MB1 and MB2. The detail of all LULCC dataset
assessed in this research are explained in the Table 1.

## 2.2 ELUC Model and Product Comparison

This study utilizes the JULES-ES model (The Joint UK Land Environment Simulator – Earth System)(Sellar et al., 2019).
JULES-ES, developed from the Met Office Surface Exchange Scheme (MOSES). JULES-ES can be used as a stand-alone
land surface model driven by observed forcing data, or coupled to an atmospheric global circulation model, such as the Met
Office Unified Model (UM) (Best et al., 2011). As JULES-ES is a process-based model that simulates the fluxes of carbon,
water, energy and momentum between the land surface and the atmosphere (Clark et al., 2011), it can used to analyze the
important role of the land surface in the functioning of the Earth System.
JULES-ES has a detailed representation of land surface processes and includes recent developments in surface physical
processes (Wiltshire et al., 2020), wood products (Jones et al., 2011), fire representation (Burton et al., 2019), dynamic
vegetation (Cox, 2001; Harper et al., 2018), land use and nitrogen cycle (Wiltshire et al., 2021), plant physiology and plant
functional types. It can simulate the historical evolution of the land carbon cycle under increasing atmospheric CO2
concentration and climate change (Harper et al., 2016, 2018). Moreover, it can also simulate the effect of land-use change



when forced with external LULCC datasets. For this study, we use the same setting of JULES-ES as in TRENDY v12 for GCB23 but vary the fraction of agriculture with several datasets as presented in Table 1.

Apart from looking at the effect of various LULCC datasets on ELUC estimates, we also investigate the ELUC uncertainties using different DGVMs. We analyse 18 DGVMs from TRENDYv12/GCB23, all driven with the LULCC forcing Mapbiomas
160    1.0.

In addition to the DGVMs, we use ELUC estimates for Indonesia from 3 Bookkeeping Models (BKM) that contributed to GCB23: H&C (Houghton and Castanho, 2023; Houghton and Nassikas, 2017), OSCAR (Gasser et al., 2020) and BLUE (Hansis et al., 2015). Although the DGVMs and the BKMs BLUE and OSCAR used the LULCC forcing MB1, it results in different estimation. Furthermore, these two model types have different approaches and parametrizations. BKM methods track
the variations in carbon storage within vegetation, soils, and wood products both pre- and post-LULCC by employing established growth and carbon decay rates over time. Unlike DGVMs, BKMs do not account for the influence of changing environmental factors on vegetation growth rates. Instead of simulating carbon stocks, BKMs rely on direct observational data for carbon densities, which are derived from biome-level values found in literature and inventory data(Houghton et al., 1983). We also compare the ELUC estimates from DGVMs and BKMs with the National Greenhouse Gas Inventory (NGHGI) from
Indonesia, a comprehensive report on greenhouse gas (GHG) emissions and removals compiled by the Indonesian government. The Indonesian NGHGI follows the IPCC (Intergovernmental Panel on Climate Change) definition of land-use emissions, which differs from the ELUC definition used by the GCB, as the NGHGI uses terrestrial fluxes occurring on all land that countries define as "managed". We specifically use data from the NGHGI Biennial Update Reports 2 and 3 (BUR2 and BUR3) (Anwar et al., 2021), which cover the time periods relevant for our study, namely 2000-2016 and 2000-2019, respectively.
BUR is a periodic national report that follows UNFCCC (United Nations Framework Convention on Climate Change) guidelines for non-Annex I countries. The GHG Inventory quantifies LULCC emissions using Tier 1 and Tier 2 methods according to the 2006 IPCC Reporting Guidelines: Emissions are calculated by multiplying the areas of land cover change with specific emission factors for each land cover conversion. The data collections and calculations are done by ministries, bureaus, or agencies. The data are collected by each relevant government organisation, then compiled by the Ministry of
Environment and Forestry (Anwar et al., 2021). Indonesia released BUR2 in 2018 (Agung Sugardiman et al., 2018) and BUR3 in 2021 (Anwar et al., 2021). BUR3 uses an improved land cover map by reinterpretation land use based on Landsat imagery and revision of relevant sector data. This creates different estimates for LULCC emissions between BUR2 and BUR3 (Anwar et al., 2021).

## 2.3 Analysis

### 2.3.1 Annual and Spatial LULCC

Quantifying the difference in products of LULCC is the first step analysis in this research. We compare the temporal changes in LULCC from the datasets LUH2- GCB2022, MB1 and MB2. MB1 classifies Agriculture as combination of Oil Palm and



Other Agriculture. MB2 has two additional classifications, namely Rice Paddy and Pulpwood Plantation. FAO classifies Cropland as combination of Arable Land and Land Under Permanent Crops (including Oil Palms). In the following, we refer

to the Mapbiomas category "Agriculture" as "cropland" to simplify the comparison across datasets.

The spatial maps of cropland area are plotted and analysed to assess the difference between LULCC datasets. We plot LUH2-GCB2022, whose cropland and grazing land areas are from HYDE based on FAO country statistics. The Mapbiomas dataset is satellite-based; thus, it is already spatially distributed. Yet, the MB1 and MB2 datasets needed to be processed as follows: First reclassify - Palm Oil, Rice Paddy, Pulpwood Plantation and Other Agriculture are defined as 100% crop while other

classifications (Forest, Non-vegetated Area and Water Body) are considered as 0% crops. The spatial resolution of Mapbiomas is 30 m, while LUH2 (used as GCB input) has a spatial resolution of 0.5 degree. We thus regridded the land cover fractions of MB1 and MB2 to 0.5 degree resolution.

### 2.3.2 ELUC Simulations

The LUH2-GCB2022, MB1 and MB2 datasets used as input to JULES-ES, which is run at 0.5 degree resolution. This

resolution allows to catch the geographical conditions of Indonesia, which is an archipelago consisting of many islands with considerable topographic variability. To calculate ELUC, we perform three sets of simulations with JULES-ES using cropland areas from the three LULCC datasets. The simulations are performed according to the TRENDY-v12 configuration, using time varying climate, $CO_2$ and nitrogen deposition following the TRENDY protocol (Sitch et al., 2024). The first simulation uses LUH2-GCB2022 over the period 1700-2022. The second and third simulations use LUH2-GCB2022 from 1700 to 1999 and

Mapbiomas (MB1 and MB2) for 2000-2022. JULES-ES is spun-up to steady state conditions by running 50 spin-up cycles of 20 years starting 1700 and then each set of simulation follows two scenarios, S2 and S3. The scenario S2 uses time-varying $CO_2$ and climate forcing but constant present-day land use (from the year 2000). The scenario S3 uses time-varying $CO_2$, climate forcing, land-use varying from 1700 to 2022 (but use 3 different datasets (LUH2-GCB2022, MB1 and MB2) from 2000). ELUC is calculated as the difference in net biome productivity at a grid-cell level between these two simulations. The

ELUC from these simulations is compared to understand how differences in LULCC from LUH2-GCB2022, MB1 and MB2 impacts the emission of carbon.

We also plot annual timeseries of all the available result from DGVM, BKM and NGHGI. We use the ensemble mean of TRENDY v12 for DGVMs; the model BLUE as BKM; and NGHGI data from BUR3. We plot the net carbon emissions to analyse their trend, magnitude and pattern. Then, we re-plot the ensemble DGVM, BKM (BLUE, H&C and OSCAR) and

NGHGI (BUR2 and BUR3) by including the peat fire emissions and peat drainage emissions. The NGHGI reports data on peat fire emissions and peat drainage emissions, whereas for DGVMs and BKMs peat fire emissions and peat drainage emissions are added from external dataset. Peat fire emissions are taken from the Global Fire Emission Database (GFED4s;Van Der Werf et al. 2017), while peat drainage emissions are a combination from three spatially explicit datasets (FAO peat drainage emissions for 1990-2020 (Conchedda and Tubiello, 2020); peat drainage emissions for 1700-2010 from simulations with the



DGVM ORCHIDEE-PEAT(Qiu et al., 2021); peat drainage emissions for 1701-2021 from simulations with the DGVM LPX-
Bern v1.5 (Lienert and Joos, 2018)). We investigate their ELUC trend, magnitude and pattern with or without peat emission.

## 3 Results

### 3.1 Annual land-use changes in Indonesia

In this section, we analyse the cropland dynamics over Indonesia for the last two decades. There are 3 datasets, LUH2-
GCB2022, MB1 and MB2. All datasets show that cropland is increasing between 2000-2022, but the total change varies
between 10 to 20 MHa (Fig. 2a). The highest increase is LUH2-GCB2022, mainly in 2016-2022 when it grows 10 MHa. The
difference between datasets can be seen clearly from the dynamics of the cropland change (Fig. 2b). LUH2-GCB2022 show
higher cropland changes than MB1 or MB2. In certain years, it shows a gain of nearly 3 million hectares in cropland, such as
in 2003 and 2016 while it shows a significant loss of over 1 million hectares in 2005, which is suspicious as we could not find
any independent evidence supporting such large decline in cropland during that year. Furthermore, the succession of 2010,
2012 and 2014 where the change in LUH2-GCB2022 cropland is exactly zero seems also is unlikely as Indonesia experienced
a continuous increased development of agriculture during the last decades (Gandharum et al., 2022; Maheng et al., 2021).
Other than that, MB1 shows a decrease in cropland area in 2019 but this year has low confidence for MB1 as 2019 is the final
year of the dataset and the Mapbiomas methodology needs validation from a year before and one year after (Mapbiomas
Indonesia - Collection).This is confirmed in MB2 that shows no decrease of cropland in 2019.
Annual cropland changes in MB2 increased from 2000 to 2011/2012 with a peak of about 1.2 MHa cropland expansion and
decreased slowly thereafter. This aligns with other research (Gaveau et al., 2022) that show forest conversion to agriculture
(specifically oil palm) increased during the 2000s, peaked around 2009-2012, and steadily declined thereafter. Also, MB2 does
not show peculiar years where cropland areas changes would be exactly zero. Overall, we expect MB2 to be the most accurate
representation of croplands we assess in this study.







**Figure 2:** **The comparison of data drivers timeseries of (a) cropland areas (MHa); (b) annual cropland area change (MHa); spatial distribution of cropland fraction in (c)LUH2-GCB2022 in 2000; (d) LUH2-GCB2022 in 2018; (e) LUH2-GCB2022 change between 2000 and 2018; (f) MB1 in 2000; (g) MB1 in 2018; (h) MB1 change between 2000 and 2018; (i) MB2 in 2000; (j) MB2 in 2018; (k) MB2 change between 2000 and 2018.**

The differences in cropland change in Indonesia across the datasets can also be observed from its spatial distribution. All 3

datasets, LUH2-GCB2022, MB1 and MB2 agree that most cropland areas are concentrated in three main islands: Sumatra, Kalimantan, and Java, with smaller regions found in Bali, Nusa Tenggara, Sulawesi, Papua and Maluku Islands (Fig. 2 c-k).





When examining the spatial distribution of cropland total, MB1 and MB2 shows higher cropland fractions in Sumatra, Kalimantan, and Java compared to the LUH2- GCB2022 dataset in the year 2000 and in 2018 (Fig. 2c, 2d, 2f, 2g, 2i, 2j).

The LUH2-GCB2022 dataset shows a significantly larger increase in cropland fractions compared to Mapbiomas in Sumatra and Java over the 2 decades (Fig. 2e, 2h, 2k). This Mapbiomas difference with LUH2-GCB2022 align with other research that show Kalimantan and Sumatra has the highest rate of deforestation before 2000 (since 1950) (Gaveau et al., 2022; Santoro et al., 2023). Then deforestation in Java and Bali already occurred before 1950 (Gaveau et al., 2022; Santoro et al., 2023). It means that all 3 main islands already had large cropland area before 2000. There are also large differences in the border regions of Kalimantan and Sumatra. These areas are known for their dense forests and have experienced major forest fires during El

Niño events in the last 2 decades, 2000-2020 (Brasika, 2023; Fanin and Van Der Werf, 2017). Moreover, these two islands contain vast areas of peatland (Anda et al., 2021; Brasika, 2022).

However, this does not mean that Mapbiomas is less uncertain than LUH2-GCB2022. For instance, MB1 and MB2 have different spatial distributions to some extent, although they both come from the same source with different versions. For example, in Sumatra 2000, the MB1 show that high cropland fraction is distributed across island (Fig 2f), while MB2 is more

concentrated in south part of Sumatra (Fig. 2i). This is also easily recognized in Kalimantan (Fig. 2h and 2k).

## 3.2 Annual model ELUC estimation

There are two sources of uncertainties in determining the ELUC magnitude and trend in Indonesia. First, the driving LULCC datasets and second the fate of the different carbon reservoirs (vegetation, litter, soils) after LULCC. We first investigate the uncertainty from LULCC by performing simulations with the different various cropland datasets (LUH2-GCB2022, MB1 and

MB2) in the same DGVM, JULES-ES (Fig. 3a).

Unsurprisingly, JULES-ES driven by LUH2-GCB2022 shows the largest interannual variability in ELUC, consistent with the changes in cropland areas in that forcing (Fig. 2b), showing higher emissions, more than 0.15 PgC/yr, in 2004 and 2017, following the largest cropland changes that happened the year before in the LUH2-GCB2022 dataset. The ELUC mean and internal variability from LUH2-GCB2022 is 0.11 ± 0.03 PgC/yr. These patterns do not appear in JULES-ES when forcing by

either MB1 or MB2, with simulated ELUC showing less year-to-year variability and being 0.08 ± 0.01 PgC/yr and 0.06 ± 0.01 PgC/yr respectively for the last 2 decades. However, notable differences also exist between ELUC driven by MB1 and MB2, following the cropland patterns shown in Fig 2. The MB2 ELUC is lower than the MB1 as cropland expansion is generally lower in MB2 (Fig. 2b). The average estimation of Indonesian ELUC of JULES-ES from different driving LULCC is 0.08 [0.06-0.11] PgC/yr.






**Figure 3:** The ELUC Model comparison (a) JULES-ES with vary drivers + ensemble Trendy v12; (b) various DGVMs in TRENDY-v12 using MB1 as driver; spatial decadal change of (c) JULES-ES (LUH2-GCB2022) 2000-2009; (d) JULES-ES (MB1) 2000-2009; (e) JULES-ES (MB2) 2000-2009; (f) JULES-ES (LUH2-GCB2022) 2010-2019; (g) JULES-ES (MB1) 2010-2019; (h) JULES-ES (MB2) 2010-2019

We now turn to the second source of uncertainty which comes from the representation of ELUC dynamic in models. This is illustrated in Figure 3b that shows 18 DGVMs (including JULES-ES) from GCB 2023. All DGVMs have the same cropland input MB1, similar with GCB 2023. The result shows that there is little agreement on the magnitude and year to year variability of ELUC across the models. The ELUC estimations from various carbon reservoirs models in TRENDY v12 is $0.12 \pm 0.02$ PgC/yr ($\alpha = 0.05$). Moreover, there is a large range from -0.04 to 0.35 PgC/yr, much higher with the range of ELUC estimation from various drivers. Thus, the various models result on higher uncertainty compared to various drivers.



In addition, we can see the impact of different driver data from its spatial distribution. The higher croplands change in LUH2-GCB2022 (Fig. 2e) results in higher estimation of ELUC is mainly located in Kalimantan, Sumatra and Java (Fig. 3c and 3f). The JULES-ES simulations based on input from MB1 and MB2 are more similar, but still show some differences in the fine

spatial details (Fig. 3d, 3e, 3g, 3h). For example, in Sumatra, JULES-ES (MB2) shows larger carbon emissions in the center of the island, while the JULES-ES (MB1) simulation has emissions spread more evenly across the island.

### 3.3 Comparison ELUC with other products

In addition to analysing the uncertainties caused by land use change driver and by the land cover change dynamics in DGVM, we also compare our findings with other estimates such as those from bookkeeping models (BKM) and those from National

Greenhouse Gases Inventories (NGHGI). From the Global carbon Budget 2023, there are 3 BKMs, hence comparable to the DGVMs analysed above. For the NGHGI, there are Biennial Update Report (BUR) 2 and 3, released by the Ministry of Environment and Forestry, Republic of Indonesia covering the 2000-2016 and 2000-2019 periods, respectively.

The result shows that ensemble TRENDY v12 ELUC mean and internal variability is similar with Bookkeeping Model BLUE ELUC estimation, with $0.12 \pm 0.02$ PgC/yr and $0.12 \pm 0.02$ PgC/yr respectively, both showing steady trend (Fig. 4a). Other

BKM H&C and OSCAR result $0.14 \pm 0.05$ PgC/yr and $0.15 \pm 0.03$ PgC/yr. The BKMs mean is 0.14 [0.12 to 0.15] PgC/yr. While NGHGI BUR3 estimates lower ELUC mean and internal variability of $0.06 \pm 0.06$ PgC/yr, with increasing trend. Furthermore, BUR3 has different annual pattern.

This emission is much larger when including the peat emissions. The carbon emissions is doubled; ensemble TRENDY v12 = $0.23 \pm 0.05$ PgC/yr; BLUE = $0.22 \pm 0.06$ PgC/yr and BUR3 = $0.13 \pm 0.10$ PgC/yr. This is consistent with other BKM such as

H&C and OSCAR with estimation $0.25\pm0.07$ PgC/yr, and $0.25 \pm 0.06$ PgC/yr. Also, other NGHGI, BUR2 estimates $0.18\pm0.07$ PgC/yr of the carbon emissions total with peat. The distinct different is also shown in all DGVM and BKM has steady trend while NGHGI tend to have increasing trend and higher variability. We also found that all models and products show the same peaks in years 2002, 2006, 2009, 2015 and 2019 (Fig. 4b). It does not appear in the emissions without peat (Fig. 4a). This is the exact same year of moderate-high El Nino.




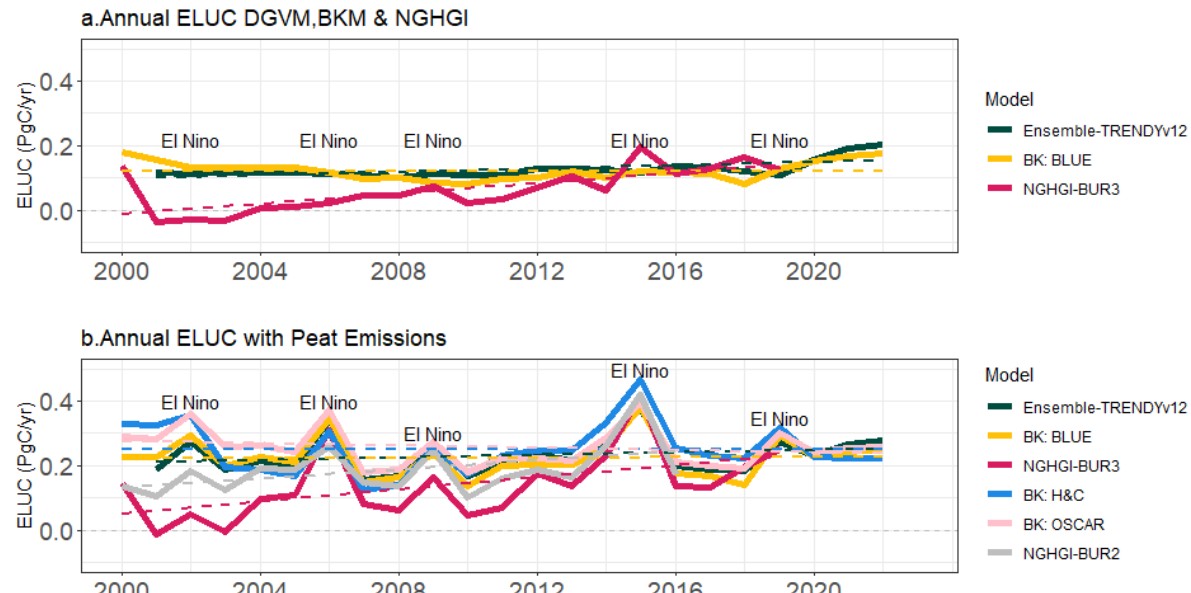

**Figure 4: ELUC comparison of different models over the period 2000-2022 (a) Annual ELUC from DGVMs, BKMs and NGHGI; (b) Annual ELUC from BLUE and BUR3 models with and without peat emissions + Ensemble TRENDY v12**


## 4 Discussion

This research analyses land cover products that are used to estimate Indonesia ELUC fluxes. The LULCC driving dataset is one of the main sources of uncertainty in ELUC trends in Indonesia (Bastos et al., 2020; Gasser et al., 2020). Thus, we compare census-based dataset of FAO, which is used widely for ELUC including Global Carbon Budget until 2022 version, and new

satellite-based datasets Mapbiomas Indonesia 1.0, which is utilised as driver dataset in GCB 2023. In addition, we also investigate the latest version of Mapbiomas Indonesia 2.0, which has not yet been used in any GCB assessments. Results show that all datasets show an increase in cropland area over the period 2000-2022, but the magnitude and year-to-year change shows limited agreement across datasets. This greatly impacts the simulated annual emissions of ELUC using our carbon cycle model, JULES.

Many models/simulations use change in agricultural areas based on statistical data reported by countries to FAO as a forcing for tree-cover loss. However, FAO agricultural statistics has high uncertainties (Ajaz, 2016; Desiere et al., 2016). By comparing the satellite-based estimates of land cover changes to those based on FAO statistics, we improve our understanding of this cropland change and its uncertainties that play key roles on carbon emissions from LULCC.

The importance of cropland transition is essential. For example, in south Sumatra where census-based (LUH2-GCB2022) and

satellite-based (MB1, MB2) show distinct differences, LUH2-GCB2022 show much lower cropland area compared to



Mapbiomas in 2000 (Fig. 2c, 2g, 2i). In this region, forest cover was shrinking rapidly between 1996-2000(Purnomo et al., 2023). However, this was not directly converted into cropland, it became shrublands in 2000, then converted into pulpwood plantations and oil palm from 2000 onwards(Gaveau et al., 2007; Purnomo et al., 2023). This process might cause different classification between satellite and census. Furthermore, it can also be seen in Java Island, where census-based dataset shows

higher increase of cropland change, meanwhile Java has been experiencing massive infrastructure development that limits the potential expansion of agriculture in this period (Gandharum et al., 2022; Maheng et al., 2021). The satellite-based has more consistency with cropland expansion in Indonesia which is related to other LULCC like deforestation and urbanization.

Apart from many potential advantages of satellite-based dataset, there is an issue with Mapbiomas Indonesia, as it lacks grazing lands in its land-use categories. It is categorized as part of the "other agriculture". Thus, in our current model, we use grazing

lands from LUH2-GCB2022 dataset as input for all simulations in JULES-ES. This might cause some double accounting of grazing land in our satellite-based simulations where grazing land might be accounted as both cropland and grazing land. However, this effect should be very small as grazing lands covers a small area in Indonesia (around 3 MHa and are considerably steady for the last two decades). Also, in JULES-ES simulation, there is no large difference regarding the carbon emissions caused by forest conversion to cropland or grazing land category.

Furthermore, another potential misrepresentation in our model is ELUC from palm oil. As palm oil has significant contributions to the forest-agricultural dynamic over Indonesia (Cisneros et al., 2021; Gaveau et al., 2021; Tsujino et al., 2016), it should be represented in the model as a specific plant functional type (PFT). Unfortunately, we currently categorize the palm oil (together with rice paddy, pulpwood plantation and other agriculture) as cropland, because there is no PFT for palm oil in our model. The lack of a palm oil vegetation type is likely to cause an overestimation of ELUC as forest to palm oil conversion

is expected to lead to lower net carbon loss than forest to non-woody cropland. As Mapbiomas Indonesia has palm oil in Indonesian land-cover (Mapbiomas Indonesia - Collection), future model simulations should consider this separation.

Our simulations can also be compared with other products such as national greenhouse gases inventories (NGHGI) and Bookkeeping Models (BKM). NGHGI is a comprehensive report compiled by a country for greenhouse gases (GHG) emissions and removals. It follows IPCC (Intergovernmental Panel on Climate Change) definitions that differ from ELUC

definition, they use terrestrial fluxes occurring on all land that countries define as managed. However, we can compare the trend of both estimations to see its consistency. The most distinct temporal pattern of census-based simulations (JULES-ES: LUH2-GCB2022) is increasing emissions in 2003-2004 and 2017, this does not appear in any other products (Fig. 4a), even compared to with peat estimations (Fig. 4b). This is also supported by other 5 NGHGI between 2001-2012 (FREL 2014, BPREDD+ 2015, INCAS 2015, NDC 2015 and FREL 2016)(Austin et al., 2018), they confirm no peaks on those years.

In addition, the ELUC simulations using satellite-based Mapbiomas have also similar spatial patterns with other research which utilize top-down approach, atmospheric inversions model. The atmospheric inversion model uses vertical profile of atmospheric CO2 concentrations (xCO2), instead of actual carbon emissions. By creating an annual average of daily xCO2 data over Indonesia from the OCO-2 satellite, it results in the spatial distribution of xCO2 anomaly (2015-2022) is highly concentrated in the middle of Sumatra and Kalimantan around 5-10 ppm per regency (Nahas et al., 2022). Furthermore,





atmospheric inversion shows carbon sinks in south Sumatra, Kalimantan and Papua. This is potentially caused by the regrowth
process after fire event, as these areas are the main burnt area for the last two decades in Indonesia (Chen et al., 2023; Van
Wees et al., 2022). However, this does not appear in our simulations, because they are measuring different thing, our model
estimates ELUC while Atmospheric Inversion indicates net flux of carbon into the land.

Another interesting part is how these products might reveal the connection between carbon emissions and climate dynamics.
All DGVMs, BKM and NGHGI with peat emissions shows strong relation with climate events which shows the peak of carbon
emissions is exactly in the same year of moderate-to-strong El Nino. This is mainly contributed by the carbon emissions from
peat fire and peat drainage. However, the emissions from land use change (ELUC) have almost zero connection with this El
Nino dynamics, it is confirmed in all products of DGVM, BKM and NGHGI without peat emissions. The carbon emissions
from peat have a key role in Indonesia, with fluxes 1-3 times higher than ELUC and stronger in El Nino years. Those should
not be neglected in future simulations.

**5 Conclusion**

Both census-based and satellite-based dataset have their own strengths and weaknesses, census-based has lower confidence as
it has no spatial distribution while satellite-based has limited temporal availability. Addressing these differences might improve
our understanding of uncertainties in carbon emissions estimates caused by the drivers. However improving the driver dataset
is not the only issue, the model itself has the potential uncertainty as each of them are created with different approaches. This
can be seen by how all DGVMs in TRENDY v12 have the same input drivers but results in a different pattern and magnitude
of carbon emissions simulations. The difference parameterisation carbon dynamics of models creates more uncertainty
compare the simulation associated with various forcing datasets. Our best estimation of Indonesian ELUC for the last two
decades is $0.12 \pm 0.02$ PgC/yr. As this is the mean of all DGVMs that also consistent with BKMs. This emission is doubled to
$0.23 \pm 0.05$ PgC/yr, if we include the peat fire and peat drainage emissions.

We find some important issues that should be addressed carefully to reduce these uncertainties in the future. First, the changing
of one land use should affect other land use. For example, the cropland change has strong connections with changing forest
and urbanization. This can be done by combination census-based and satellite-based datasets. Second, inclusions of peat
emissions and climate effect (El Nino) in the models, as this factor has major contributions in carbon emissions in Indonesia.
Third, palm oil representations which are dominant in Indonesia's agriculture with a distinct carbon cycle but have been
simplified here.

Apart from many uncertainties created by the drivers and models, and some representations such as peat emissions, palm oil,
and climate dynamics. All products agree that Indonesian carbon emissions from LULCC have had no decreasing trend for the
last two decades. Furthermore, NGHGI shows increasing trends. Thus, it seems that prevention of deforestation and fire had
low impact in Indonesia on reducing carbon emissions for addressing climate change.





## 6 Code Availability

The simulation can be accessed through reasonable request to the correspondence author.

## 7 Data Availability

Mapbiomas data is freely available at https://mapbiomas.nusantara.earth . LUH2 can be accessed on https://luh.umd.edu . All
TRENDY data are freely available from the following website: https://globalcarbonbudgetdata.org/.

## 8 Author Contribution

IBMB, PF and SS designed the ideas and methodology. MOS develop ELUC simulation on JULES-ES and its analysis tools.
TMR develop approach on Mapbiomas data gathering and preprocess. MCD set-up JULES-ES. KKG prepared HYDE, include
the extrapolation of Mapbiomas. JP and CS prepapred Bookkeeping Models and its analysis. GH and LC create the LUH2-
GCB2022 and also the LUH2 for TRENDY v12 simulations. IBMB analyzed the data and results discussed with all authors.
IBMB and PF led the writing of the manuscript to which all authors contributed.

## 9 Competing Interests

This research has been supported  by Indonesian Endowment Fund, or known as LPDP (Lembaga Pengelola Dana Pendidikan).
As part of PhD scholarship that is received by IBMB.

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
