# Peer review of "Uncertainties in carbon emissions from land use and land cover change in Indonesia"

_EGUsphere, 2024_

## Community Comment (CC1)

[revised manuscript text omitted]

---

## Author Response (AR1)

Referee Comment 1:

In their study, the authors investigated the uncertainties in carbon emissions from land use and land cover change in Indonesia. They analyzed spatiotemporal differences in cropland from several LULCC datasets. The authors run multiple simulations to disentangle the impact of different LULCC datasets on the carbon emissions. They also compared the results of different LULCC data-driven JULES-ES models and the results of MB1-driven TRENDYv12 simulations, as well as comparing their estimates with other models/products such as the National Greenhouse Gas Inventory of Indonesia (NGHGI). Overall, testing the sensitivity of the model to the LULCC changes is of great interest, especially for some important regional carbon emission assessments, and the manuscript is decently written.

However, I have some concerns as follows: (1) In the Introduction, the authors use a lot of paragraphs to describe the importance of Indonesia in the global carbon cycle, which is very important background knowledge, but this part should be streamlined. Even, the authors spend a lot of space on the introduction of some commonly used data, such as LUH2. It is unnecessary to introduce these too much, because it can appear in the data description section. Particularly, there is no review of similar studies and comparison with previous studies. Moreover, too much speculation appears in the Discussions and Conclusions, while there are no results to support them. Some results need to be explained better.

I have reduced several sentences in the introduction, include the explanation about LUH2. I also added the review similar studies in paragraph 5 of introduction.

I have deleted several sentences in discussion and conclusion to address the speculation.

(2) The innovativeness of the present version cannot be observed and was not highlighted in the Introduction.

I have added a sentence about the innovation in the introduction:

"However, we emphasize more on the uncertainties, that is not only cause by the input but also the different approach of models and/or inventories."

As this research prove that the uncertainties is not only from the different input, but also amongst model itself, as different DGVMs estimates differently with the same input.

(3) I am very concerned about the evaluation of the model as it directly determines the reliability of this study. However, the authors did not provide details on model parameter and their settings, which modules were used, and how carbon emissions were calculated. It is unclear. In particular, the authors used 0.5° resolution land use data (resampling 30 m Mapbiomas data to 0.5 degrees), only about 760 pixels in Indonesia, to run the model and analyze the spatial and temporal variability of the LULCC and investigate their impact on carbon emissions, which will lead to a great deal of uncertainty, and this make me and the readers very sceptical about the reliability of the study.

See new supplementary section JULES-ES model description which show we calculated the carbon emissions especially how land-use change affect it.

The 0.5 resolution is chosen so it can be comparable with the GCB and Trendy output. As one of the datasets is LUH2-GCB22 and all other DGVMs has the resolution of 0.5, so we use the same resolution for running the Mapbiomas.

Furthermore, this resampling process does not massively change the total area. For example, , cropland on MB2 in 2000 is 38.51046 Mha for 30 meters resolution but it is 39.51675 Mha in 0.5 degree resolution.

(4) It seems that this study only focuses on cropland changes, while the title suggests that the authors are concerned about the impact of LULCC changes on uncertainties in carbon emission. Moreover, it should be noted that LULCC not only includes changes in the internal planting structure of a specific land use type (e.g., the area of cropland remaining unchanged, but its planting type can change, thereby affecting carbon emissions), but also includes the mutual transformation between different land use types (e.g., the conversion between cropland and forests). However, this study seems to primarily focus on changes in the area of different land use types.

Thanks a lot, about this input. This research is mainly focused on carbon emissions from land use change, which is mainly related with the deforestation. There are many forested area converted to other land use change, not only cropland, but also other agriculture, and palm oil contribution is increasing the last 20 years. However, as there is no PFT for palm oil, there are some simplifications we should do, but at least we remain capture the effect of this deforestation at the first degree to the carbon emissions. Furthermore, as the Mapbiomas is derived from satellite directly, the land use change should be related with the land cover too. The changing of land cover determines the classification of land use.

Further detailed comments follow below.

**Abstract:**

Line 39:  Please keep the expression of 0.24 [0.22-0.25] Pg C/yr consistent with that of 0.23 ± 0.05 Pg C/yr. Make them comparable.

Done, thanks.

Line 44: Change the "," to ".".

Done, thanks.

**Introduction**

Line 47: I suggest adding the relevant references to support this statement.

Done.

Line 55: What does MHa mean? Is it Mha?

Yes Mha. I fixed it now.

Line 65: Delete "(Emissions of Land Used Change)" as already mentioned above.

Done.

Lines 80-82: Add the references.

Done.

**Methods:**

Line 105: It appears that only cropland was considered in this study, but not grazing land. This is confusing.

It is the standard input for JULES-ES and LUH2 that needs both cropland and grazing land. Those are both considered. Unfortunately, as Mapbiomas has no specific grazing land category, we instead used the grazing land from LUH2. This relation can be seen in Figure 1.

Although all JULES simulation uses the same grazing land dataset from LUH2, there is adjustment of grazing land on simulation with Mapbiomas when the total fraction is more than 1, so this should be explained.

Line 107: Which method have you used to reduce the grazing land fraction? Please describe the method in detail and add one or two sentences here.

I have explained it on lines 105-107 and it is simple method. Basically we sum the cropland fraction and grazing land fractions, and if the total fraction is greater than 1, we reduce the grazing land fraction so the maximum fraction of cropland+grazing land is 1.

I added the following sentence to make it clearer:

"In case total combination of Mapbiomas agriculture fraction and LUH2- GCB2022 grazing land fraction is higher than 1 in a given grid-cell, we reduce the grazing land fraction accordingly. So the total area of cropland + grazing land will be equal or less than 1."

Line 131: We could not get some detailed information about these data from Table 1, and I suggest adding some basic parameters about these data, etc.

I added information about these data in table.

Line 145: I am not an expert on the JULES-ES model. For a reader who may not know the JULES-ES model. Please add some important information on this model to the Supplementary, for example, adding the important parameters used in your study and their settings for different plant functional types (PFT), and the equations for the main modules (e.g., how do you calculate carbon emissions?) that used in your study, etc., making it so much easier to understand the scope of the model quickly and be able to interpret your study.

I have created a new supplementary to answer this. Thanks for the suggestions.

Line 190: But there is a big difference in the definitions of agriculture and cropland. In particular, they may have different parameter settings in the model. Generally, "cropland" is commonly used in the models (e.g., JULES-ES model). Are pulpwood or palm oil plantations croplands? These may affect your results and conclusions.

At the moment we treat palm oil and pulpwood as cropland. This is a simplification, but as our focus in on land use change, not land-use, our method will capture the first-order impacts of deforestation/reforestation events.

We do understand that this might affect the uncertainties. But it also results uncertainties if we represent Palm Oil/pulpwood as other PFT. For example, if we classify the palm oil and/or pulpwood as forest, it will not capture the first-order of deforestation and create uncertainties too. Thus, we address it in the discussion and conclusion.

Line 200: We know that Indonesia is mainly composed of islands, and with such a coarse resolution (0.5 degrees) used in this region, it would introduce very large uncertainties. The 0.5-degree resolution is commonly used in global-scale simulations.

This resolution is chosen so it is comparable with the resolution of GCB and Trendy, which is used as global estimation with national details. This is also the resolution of the global climate forcing dataset, CRUJRA. We use LUH2-GCB22 as our basis, and regrid Mapbiomas to the same resolution so it can be comparable.

Indeed it may result in more uncertainties, however the vegetation in the grid is represented as fractions of the grid cell. For example, the total cropland inside the grid is the same/similar for the 0.5 degree or 30 meters resolution. But the exact location of cropland inside the grid might not well represented.

You can check the number and figure bellow under your review for line 241.

Line 221: How to calculate the ELUC trend, magnitude, and pattern with or without peat emission? It's not clear.

Done. Fixed it.

**Results:**

Line 227: Replace "million hectares" with the same unit, Mha.

Done.

Lines 237-238: This sentence "This aligns with other research…" seems to belong in the Discussion section. I suggest moving it to that section.

Done.

Line 241: In Fig. 2, are the results of spatial and temporal variations in cropland all based on 0.5 degrees? I'm curious how the cropland areas of MB1 and MB2 change at their original resolution? The same or different? This may affect the reliability of all the results. Besides, the scale bar is missing in Figure 2.

Yes all are at the same 0.5 degrees resolution to can comparable.

For example, cropland on MB2 in 2000 is 38.51046 Mha for 30 meters resolution but it is 39.51675 Mha in 0.5 degree resolution. In 2018, 47.43622 Mha for 30 meters, and 48.49098 Mha in 0.5 degree resolution. Here I show you the original Mapbiomas.

[Figure]

The scale bar is not missing. The first scale bar is for 2c,d,f,g,l,j, the second scale bar is for 2e,h,k.

How are Figure 2e, 2h, and 2k calculated? 2018 minus 2000 or 2000 minus 2018? Please explain in the figure caption.

Done. It is 2018 minus 2000.

Lines 250-256: Part of the discussion?

Done.

Line 294: "0.12 ± 0.02 PgC/yr and 0.12 ± 0.02 PgC/yr respectively"? Please check these data carefully. They are not the same in Fig 4a.

They are not the same, but similar. If we continue the decimal, it will show the difference. But here I only put until 2 decimals, so it consistent with others.

**Discussion:**

Lines 308-309: It seems that you are not comparing it to the FAO data. I checked the entire results section and found no comparisons to the FAO data. Why? But you mentioned that data in above text and conclusions. Please add relevant results.

Yes, you are correct. I fix it.

I mean the LUH2 that the cropland derived from FAO, not the FAO dataset directly.

Line 314: JULES or JULES-ES?

JULES-ES. I fix it.

Line 328: Don't repeat what you've already mentioned above (see Line 105).

Done, fixed it.

Lines 333-334: There are no results to support this, it's plain speculation. Similar speculation comes up too many times in your research.

This sentence does not based on the results, this is based on how the JULES-ES treat cropland and grazing land. I added a reference about it.

Lines 337-338: I doubt that it is correct to classify palm oil as cropland? Would the simulation results change if it was classified as forest or other land cover types? This leads to a great deal of uncertainty in your results. Please show the results of these differences.

Yes, the palm oil should not classify as cropland, but it also should not classify as forest. In other hand, this category should not be neglected as it has high percentage. Unfortunately, there is no palm oil PFT in the models yet.

Lines 360-361: It's a very interesting result, but you don't have any more explanations. Can you discuss it in depth?

Yes, I added several sentences to discuss it:

"This is mainly contributed by the carbon emissions from peat fire and peat drainage. During El Nino year, Indonesia experiences drier and hotter climate condition, especially in the Sumatra

and Kalimantan islands (Brasika, 2021; Nurdiati et al., 2022). This condition is favourable for fire regime in the area that not only burnt the above ground biomass, but also peat soil below ground(Brasika et al., 2021; Fanin and Van Der Werf, 2017). As peat contains massive carbon and peat fire is hardly detected and managed(Indradjad et al., 2024), this results the peat fire release massive amount of carbon during hot and dry El Nino year(Stockwell et al., 2016)."

**Conclusion**

Lines 372-373: Didn't find your results to support that conclusion. Suggest removing it from the text.

Done. Removed.

**JULES-ES model description (Land-use Change impact on Carbon Cycle):**

a.  Vegetation distribution
    Land use change in JULES-ES impact on vegetation distribution by modifying the competition term on the simulation of PFT distribution. Here in Equation (1).

$$\frac{dv_i}{dt} = \frac{\lambda \prod v_*}{C_{vi}} \left\{ 1 - \alpha a_i - \sum_j c_{ij} v_j \right\} - \gamma_v v_* - \beta_i v_* \tag{1}$$

Here, $v_i$ denotes the area of grid covered by PFT $i$. The rate of change in $v_i$ depends on the carbon available for increasing the PFT area ($\lambda \prod v_*$), and the associated carbon cost, determined by the carbon density ($C_{vi}$). Four terms balance the constant expansion of PFTs:

1.  **Vegetation Loss** ($\gamma_v v_*$): Represents vegetation loss from mortality process, not related to competition
2.  **Fire disturbance** ($\beta_i v_*$): Accounts for vegetation loss due to fire
3.  **Competition amongst PFTs** ($\sum_j c_{ij} v_j$): The dominant PFT will out compete the others.
4.  **Land-use change** ($\alpha a_i$): Represent the competition caused by Land-use change

In the context of land-use change, $\alpha$ is the disturbed fraction, and $a_i$ equals 1 for non-woody PFTs and 0 for woody PFTs. Woody PFTs are restricted from growing in the disturbed fraction, while non-woody PFTs can grow anywhere, including disturbed areas, where they are considered agricultural grasses. These grasses are physiologically identical to natural grasses but are labelled differently. The value of $\alpha$ can change over time. When $\alpha$ increases, natural grasses are reclassified as agricultural grasses, and woody PFT areas are replaced first by bare soil, then potentially by non-woody PFTs if viable. Conversely, as $\alpha$ decreases, agricultural grasses are reclassified as natural grasses, and woody PFTs can re-expand into the grid.

b.  Soil carbon store
    Land-use change, together with fire, affect the soil carbon store by altering the flux of vegetation-to-soil litter. This litter flux is composed of:
    *   Local litterfall from leaf, root and stem turnover.
    *   Litter produced by disturbances and competition.

The effect of land-use change, and fire are integrated into the following equation:

$$\Lambda_{CvLoss} = \sum_i v_i \left( \Lambda_{li} + (\gamma_{vi} + \beta_i)C_{vi} + \Pi_i \sum_j (\alpha a_i + c_{ij} v_j) \right) \tag{2}$$

Here, $\Lambda_{CvLoss}$ represents vegetation carbon loss. However, not all carbon lost enters the soil carbon pools; some loss due to land-use change is diverted to wood-product carbon pools. The litter generated by land-use change ($\Lambda_{LUC}$) is calculated using the disturbed fraction ($\alpha_{-1}$) from the previous time step, as shown in Equation (3):

$$\Lambda_{LUC} = \Lambda_{CvLoss} - \sum_i v_{LUC,i} \left( \Lambda_{li} + (\gamma_{vi} + \beta_i)C_{vi} + \Pi_i \sum_j ((\alpha - \alpha_{-1})a_i + c_{ij} v_{LUC,j}) \right) \tag{3}$$

Here, $v_{LUC}$ is the PFT area computed from Equation (1) using $\alpha = \alpha_{-1}$. The litter produced by land-use change is distributed between soil carbon pools and wood-product pools. The below-ground carbon portion, determined by root carbon ($C_v$), is added to the soil carbon pool, while the remaining above-ground carbon is allocated to the wood-product pool.

Referee 2 Comment:

This study investigates the uncertainties in Indonesia's carbon emissions from land use and land cover change (LULCC). The authors analyzed spatiotemporal variations based on multiple LULCC datasets and conducted simulations to assess their impact on carbon emissions. They compared results from different LULCC data-driven JULES-ES models with MB1-driven TRENDYv12 simulations and benchmarked their estimates against other models and datasets, including Indonesia's National Greenhouse Gas Inventory (NGHGI). While satellite-based datasets help reduce uncertainties, discrepancies also stem from variations in carbon pool representations across models. The study estimates Indonesia's carbon emissions from land use change at 0.12 ± 0.02 PgC/yr with a stable trend, which doubles when peat fire and peat drainage emissions are included. Assessing datasets uncertainty and model variation to LULCC changes is crucial for improving regional carbon emission estimates, making this research a valuable contribution to the field.

The study is well-executed, and the manuscript is generally well-written. However, several key aspects require further clarification and more detailed discussion before a definitive conclusion can be drawn. I recommend a major revision to address these issues before the manuscript can be considered for publication.

Major comments:

**1 The JULES-ES model was used to simulate the annual ELUC trend and the mean ELUC spatial distribution. However, based on the material presented in the manuscript, the model's reliability remains unclear. The study does not provide a comprehensive understanding of the model's mechanisms for simulating the carbon cycle, the key factors driving its outputs, or the most influential variables in deriving the results.**

Additionally, the model's reliability is further questioned due to the lack of site-level validation, particularly for the region having different land types, site validation against all those types should be conducted. Before applying the model at a regional scale, validation against observational data is essential to ensure accuracy.

I suggest the following improvements:

1. Provide a more detailed explanation of the model mechanisms either in the method section, particularly the module responsible for calculating carbon cycles, to help readers better understand how the model functions.

2. Include site-level validation against observational data to demonstrate the model's ability to capture carbon emission trends. If such validation has been conducted previously, please cite the relevant publications and incorporate a figure or table to present the validation results.

Thank you very much for this strong input. Regarding JULES-ES model mechanism, I agree that it needs to be explained further to make clear reliability of the model. Thus, I created supplementary for my manuscript to help explaining detail mechanisms of JULES-ES, especially

how it calculates the carbon cycle. About the validation, as JULES-ES is one of DGVM that has been widely used for carbon cycle model, I will cite relevant publications as you suggested.

**2 Each model's carbon emission estimates are highly dependent on climate forcing, and variations in model sensitivity to these factors can lead to different results. However, the manuscript does not clearly identify the key climate drivers influencing carbon emission calculations. The study primarily focuses on the effects of land use change, but given that Indonesia is largely dominated by peatlands, other factors must be considered. Changes in global temperature and precipitation can significantly impact the carbon cycle in peatland ecosystems, as well as other soil systems, potentially playing a role equal to or even greater than land use change. To better understand the broader influences on ELUC, I recommend conducting a sensitivity test of the JULES-ES model against major climate forcings. This would help quantify the impact of climate change on carbon emissions in the absence of land use change, providing a more comprehensive assessment of emission drivers.**

Yes , you are correct that all of the emissions estimates are highly dependent on climate forcing. Thank you very much for this comment. All those models presented in this paper has the same climate forcing as all of them following protocol from Trendy which also part of Global Carbon Budget. This will erase the uncertainty caused by climate forcing, but the importance of the climate forcing remain essential. Furthermore, this research is mainly focus on land use and its model. I agree that peatland and climate has crucial role on Indonesia carbon emissions, this is can be seen clearly in figure 4b where the estimation increase massively and have strong connection with El Nino when including emissions from peat drainage and peat fire. I will add more comprehensive assessment of emissions drivers as you suggested.

**3  The 0.5-degree resolution used for simulating the Indonesian region is relatively coarse, which may limit the accuracy of the results. Additionally, extrapolating land use datasets from a 30m grid to 0.5-degree resolution introduces significant uncertainties and potential errors. To strengthen the study's conclusions, I recommend either conducting a regional simulation with a finer resolution or providing a more detailed discussion of this limitation and its potential impact on the findings.**

I agree that the 0.5 degree resolution is relatively coarse for country level assessment such as Indonesia. However, this resolution is chosen so it can be comparable with other models which derived from the models that has been used on global carbon budget. Apart from that, the re-griding of 30m to 0.5 degrees my reduce the certainty of spatial distribution, but it will not affect a lot of total area. As input of the models is percentage of area in each grid box, it will not change too much. For example, cropland on MB2 in 2000 is 38.51046 Mha for 30 meters resolution but it is 39.51675 Mha in 0.5 degree resolution. In 2018, 47.43622 Mha for 30 meters, and 48.49098 Mha in 0.5 degree resolution. Here I show you the original Mapbiomas.

| | MB1 Cropland 2000 | MB1 Cropland 2018 |
|---|---|---|
| |
[Figure]
 | |
| | MB2 Cropland 2000 | MB2 Cropland 2018 |

[Figure]

Specific comments:

In the method section, please consider adding more details and possibly a figure to show the key climate trend through the simulation period.

This has been explained in the text that the climate is following TRENDY protocol.

In the method section, please consider adding one figure to outline the modern geographical feature distribution of the study area including land types (agricultural and non-agricultural) to let audience know the rough picture of the study region.
This can be seen in Fig. 2.

Figure 2. The upper panel uses MHa and the lower one uses km^2. Please make them consistent.

I do not think it will be good idea. As it will make the legend unclear. Meanwhile the upper panel and lower panel has different context. Upper panel is total area of whole country, while lower panel show the detail spatial grid.

Line 261. Too assertive to assume the sources of uncertainty come from two aspects - land use datasets and model mechanisms. What about the uncertainty in climate forcing etc? Consider removing this statement.

Yes, there will be potential uncertainties that might be caused by climate forcing. But it is not considered in this research, as it is focused on land use and model. Furthermore, all the models use the same climate forcing.

Line 281. The conclusion is too subjective by sorely comparing two sources of uncertainty. Consider removing.

Done.

Figure 3b. It is hard to see the baseline model JULES used for comparison. Please highlight the trend of it.

Sure.

Figure 3c-h. It looks like most of the pixels fall within 0 - 0.1 purple area and it is hard to see the spatial variation for this range. Consider adding more color scales for the 0 - 0.1 range to further break it down.

The range 0-0.1 is the area of very small or no emissions from land use change. This is highly connected to lower panel in figure 2 which shows the area of land use change. Adding the color scale in this range might not add many information and might reduce the clarity of the figure.

Line 333 - Line 334. There is no direct evidence supporting this. Try to add more content / explanation.

I create supplementary to explain this.

---

## Referee Report (RR1)

The authors have improved the manuscript quite a lot. The discussion now contains much more explanations to bring the results into context. I still have some more remarks, and would say that revisions are still necessary.

1. In the authors' response to my general comment 2 (i.e.,(2) The innovativeness of the present version cannot be observed and was not highlighted in the Introduction.), I think the authors still did not point out the novelty of the study. Indeed, assessing model uncertainty is already common knowledge from the point of view of input data and model structure. This does not seem to be a point of innovation. The authors still need to state the innovations of this study in detail.

2. In the authors' response to my general comment 3, ok, the authors have provided some description on JULES-ES model in the revised version. Indeed, these basic descriptions can be found through the original references. However, the reader is more interested in how the authors applied the model to this study. For example, the authors did not provide details on model parameter and their settings in your study area; how to determine some important parameters relevant to this study and how the model was validated by site-based observations. Despite the fact that the authors cite relevant publiactions, this is still hard to believe the results of their study, and I strongly request the authors to provide the results of the validations based on site observations (i.e., Flux Tower Eddy-Covariance Measurements). Although JULES-ES model is one of DGVM and is widely used, the authors need to further test it using observations when applying it to Indonesia, which is directly related to the reliability of this study.

3. The authors mentioned that 'JULES-ES is one of Trendy V12 and has been widely used'. If the 0.5 degree resolution was adopted in this study, it appears that the results of this study can be obtained directly from Trendy v12. This study appears to look like repetitive work. In addition, $0.5^{\circ}$ is fine for large-scale assessments like the globe, while caution is needed when using it for regions. The authors have introduced a lot of uncertainty by resampling some of the higher resolution data (30m) to $0.5^{\circ}$, even though the authors state that it is for better comparisons. I would strongly recommend that the authors add the assessments of 30m to the supplementary to make the results more reliable.

---

## Author Response (AR2)

The authors have improved the manuscript quite a lot. The discussion now contains much more explanations to bring the results into context. I still have some more remarks, and would say that revisions are still necessary.

1. In the authors' response to my general comment 2 (i.e.,(2) The innovativeness of the present version cannot be observed and was not highlighted in the Introduction.), I think the authors still did not point out the novelty of the study. Indeed, assessing model uncertainty is already common knowledge from the point of view of input data and model structure. This does not seem to be a point of innovation. The authors still need to state the innovations of this study in detail.

   Thank you for addressing this. I have added paragraph/sentences in introduction and conclusion. Despite focusing on the uncertainties only, I provide some agreements between models/products as focus in a point of innovation.

2. In the authors' response to my general comment 3, ok, the authors have provided some description on JULES☐ES model in the revised version. Indeed, these basic descriptions can be found through the original references. However, the reader is more interested in how the authors applied the model to this study. For example, the authors did not provide details on model parameter and their settings in your study area; how to determine some important parameters relevant to this study and how the model was validated by site-based observations. Despite the fact that the authors cite relevant publiactions, this is still hard to believe the results of their study, and I strongly request the authors to provide the results of the validations based on site observations (i.e., Flux Tower Eddy-Covariance Measurements). Although JULES-ES model is one of DGVM and is widely used, the authors need to further test it using observations when applying it to Indonesia, which is directly related to the reliability of this study.

   The model description of JULES-ES has been explained in the supplementary that I attached in the previous answer-question. I will add the parameter settings that I use in this model in the supplementary as you suggest.

   I agree that validating the model with site-based observations such as EC tower is important research, but I believe this is out of scope of this research. As our main focus are addressing global/regional/country-level models/product uncertainties and find the agreement amongst them. Evaluating JULES-ES model is not our goal, apart from that JULES-ES is only one of many models we use in this study (18 DGVMS, 3 BKM & 2 NGHGI). So, there is no point focusing on evaluating one model only, as the uncertainties will remain there when compared to other models/product. We use JULES-ES in one section just to illustrate how different LULCC affect the uncertainties.

3. The authors mentioned that 'JULES-ES is one of Trendy V12 and has been widely used'. If the 0.5 degree resolution was adopted in this study, it appears that the results of this study can be obtained directly from Trendy v12. This study appears to look like repetitive work. In addition, 0.5° is fine for large-scale assessments like the globe, while caution is needed when using it for regions. The authors have introduced a lot of uncertainty by resampling some of the higher resolution data (30m) to 0.5°, even though the authors state that it is for better comparisons. I would strongly recommend that the authors add the assessments of 30m to the supplementary to make the results more reliable.

   There is no repetition in this research with Trendy v12. As our simulation use Mapbiomas Indonesia 1.0 (MB1) directly, but Trendy v12 use MB1 that has been adjusted with HYDE. We run this MB1 directly in JULES-ES to do assessment, before this new input widely utilize in

trendy. Apart from that, we also run MB2 that is not in Trendy v12. However, we use the similar setting for the land use change and PFT parameterization to make it consistent, so it reduces the potential of other uncertainties. Then we can focus to the uncertainties from the different land use change input.

Regarding the 0.5 resolution, this is not only use for global assessment only. This is also use in regional assessment, known as RECCAP (Regional Carbon Cycle Assessment and Processes). One of the RECCAP region is Southeast Asia, where the area is dominated by Indonesia.

I will add the map comparison of land use in the supplementary, similar with my previous answer-question. However, running the JULES-ES or other models in the 30 meters seems unnecessary to be implemented as we do not have climate forcing with resolution of 30 meters. Even if the 30 meters climate forcing is available, the running with this will result in more uncertainty as there is uncertainty caused by different climate forcing. This will result the research broader and not focus to the main goal.